# The Influence of Social Dynamics on Biological Aging and the Health of Historically Marginalized Populations: A Biopsychosocial Model for Health Disparities

**DOI:** 10.3390/ijerph21050554

**Published:** 2024-04-26

**Authors:** Lok Ming Tam, Kristin Hocker, Tamala David, Edith Marie Williams

**Affiliations:** 1Department of Environmental Medicine, School of Medicine and Dentistry, University of Rochester, Rochester, NY 14642, USA; lokming_tam@urmc.rochester.edu; 2Clinical and Translational Science Institute, University of Rochester Medical Center, Rochester, NY 14642, USA; tamala_david@urmc.rochester.edu; 3School of Nursing, University of Rochester, Rochester, NY 14642, USA; kristin_hocker@urmc.rochester.edu; 4Department of Nursing, State University of New York Brockport, Brockport, NY 14420, USA; 5Office of Health Equity Research, School of Medicine and Dentistry, University of Rochester, Rochester, NY 14642, USA; 6Center for Community Health and Prevention, University of Rochester, 46 Prince St Ste 1001, Rochester, NY 14607, USA

**Keywords:** social dynamics, aging, health disparity

## Abstract

Historically marginalized populations are susceptible to social isolation resulting from their unique social dynamics; thus, they incur a higher risk of developing chronic diseases across the course of life. Research has suggested that the cumulative effect of aging trajectories per se, across the lifespan, determines later-in-life disease risks. Emerging evidence has shown the biopsychosocial effects of social stress and social support on one’s wellbeing in terms of inflammation. Built upon previous multidisciplinary findings, here, we provide an overarching model that explains how the social dynamics of marginalized populations shape their rate of biological aging through the inflammatory process. Under the framework of social stress and social support theories, this model aims to facilitate our understanding of the biopsychosocial impacts of social dynamics on the wellbeing of historically marginalized individuals, with a special emphasis on biological aging. We leverage this model to advance our mechanistic understanding of the health disparity observed in historically marginalized populations and inform future remediation strategies.

## 1. Introduction

Biological aging involves a progressive decline in organismal integrity and functionality via a stochastic process of cellular damage accumulation over time [1], constituting the predominant risk factor for most chronic diseases. Aging can be influenced by both genetic and environmental factors. Since the completion of the human genome project in the 2000s, environmental risk factors have been found to play a significant yet underestimated role in the etiology of age-related chronic diseases. The notion that environmental factors impact the pathogenesis of chronic diseases and, potentially, the aging process was validated by the Global Burden of Disease (GBD) project and exposure–disease association studies. These environmental risk factors contribute to approximately 60% of global deaths [2,3]. In parallel with the prevailing enthusiasm of understanding the causal relationship between environmental chemical exposure and health [2,3], environmental health science also delves into delineating the detrimental effects of environmental stressors other than exposure to toxicants on wellbeing. Examples of non-toxicant exposures include inaccessibility to healthcare, inappropriate occupational practices, poor living infrastructure, and anthropogenic climate change [2,4,5]. Together, these exposures can be summarized by the paradigmatic concept of the exposome, which encompasses the totality of environmental exposure throughout the course of life (see Figure 1) [6,7]. The exposome represents the environmental component of one’s personal exposure profile (as the driver) and consequent unique personal experience (as the matrix) under the gene x epigenetic x environment interaction [8], suggesting a strong linkage between personal experience and health outcomes. Meanwhile, according to geroscience—an integrative and targeted research field related to the study of aging as the root cause of chronic diseases [9]—one’s healthspan (i.e., time without chronic diseases) and lifespan (i.e., life expectancy), as well as their likelihood of developing chronic diseases, are largely driven by their aging trajectory. Thus, there is an emerging interdisciplinary interest, from social psychology and neuroscience to environmental science and aging biology, in understanding the impact of psychosocial stressors on aging trajectory.

Social dynamics have been proposed to affect one’s cumulative stress (i.e., allostatic load) and health in a paradoxical way. For most people, social interaction is a source of protection from stress, but for others, social interaction can manifest as a great source of stress. Accumulating lines of evidence suggest that the nature and quality of one’s social dynamics act as a determinant of health [10,11,12]. It was proposed that social dynamics, and thus social wellbeing, potentially determine one’s potential to achieve successful aging by modulating the competency of fulfilling basic sociopsychological needs (e.g., autonomy, connectedness) [13,14]. Theoretically speaking, social dynamics can be divided into social stress and social support aspects, both of which are well studied domains in sociopsychology. In concordance with the Developmental Origin of Health and Disease theory in environmental health science (reviewed previously [15]), we propose that one’s personal experience and personal psychological needs are largely mutually shaped by early-life and lifelong biopsychosocial outcomes (e.g., life satisfaction, neuroendocrine) resulting from social dynamics per se. Amidst recent social movements (e.g., Black Lives Matter, Stop Asian hate) and the COVID pandemic, heightened attention has been paid to the psychosocial influence of social dynamics, consisting of discrimination [16,17] and social isolation [11,18,19,20], on human physiology and the etiology of chronic diseases. Importantly, the fundamental question of how social dynamics-associated personal experience unfolds in these physiological outcomes and aging trajectory across the lifespan remains under-acknowledged.

Here, we explore the physiological mechanisms underlying the influence of social dynamics as a social determinant of health, under the umbrella of social stress and social support theories, on biological aging (i.e., aging trajectory) in individuals from historically marginalized social backgrounds. In this review, historically marginalized populations are defined as the groups of people who have historically been oppressed, opposed by, discriminated against by, and otherwise excluded from the larger society. These populations are likely to suffer from adverse life circumstances that drive psychological distress. Examples of such circumstances might be social injustice, environmental injustice, and a lack of a political voice. However, considering the diversity of the unique struggles experienced by each member of these communities, the scope of this review will focus on developing a conceptual, instead of “one-for-all”, framework to facilitate the categorization of the social factors that impose social stress and social support in their social dynamics. By reviewing the literature across disciplines, we aim to provide a scientific framework for understanding how social dynamics impacts the psychosocial aspect of biological aging among historically marginalized populations and leverage this framework for the better enactment of public policy addressing the health disparities in these populations. By proposing an overarching biopsychosocial model, we focus the scope of this paper on the impacts of social dynamics on the biological aging of historically marginalized populations. We also discuss how this model can inform future intervention strategies to enhance the competency of successful aging among these marginalized populations.

## 2. Social Dynamics, Stress Physiology, and Biological Aging

To understand how the social experiences of historically marginalized populations influence biological aging, deciphering the “regulators and mediators” of sociopsychological stressors is indispensable. These regulators and mediators can be roughly explained by social stress and social support theories, whereby social stress and social support govern the magnitude of perceived psychological distress and cumulative stress in the human brain and body. Before discussing how social dynamics cause health disparities among historically marginalized populations with the overarching biopsychosocial model, the effect of two components of social dynamics—social stress and social support—on stress physiology and its consequential impact on aging trajectories will be summarized in the following sections (see Figure 2).

### 2.1. Social Stress Theory

Social stressors exist as forces external to a person; they can be structural, constitutional, infrastructural, conceptual, or interpersonal [20,21,22,23,24,25]. These stressors constitute sources of challenging conditions that destabilize their functional integrity psychologically and physiologically, leading to emotional distress [20,21,22,23,24,25]. In social stress theory, a person tends to experience negative emotional feelings in interpersonal relationships in which they are not equally treated by others, which trigger psychological corrective actions (e.g., venting) and physiological reactions (e.g., the release of neuroendocrine hormones) [25,26]. There are numerous factors underlying the social inequality experienced by historically marginalized populations due to race, ethnicity, physical ability, gender, and more. These include stereotyping, prejudices, stigmas, and unconscious bias, as well as conscious bias in terms of historical and institutional discrimination policies that remain unredressed. These social inequalities present as interpersonal threats, such as microaggressions (i.e., everyday discrimination), discrimination, social stigmas, historical trauma, and the corresponding inequity in accessing the societal resources needed to fulfill their basic psychological needs (e.g., employment, connection). Altogether, various forms of social inequality induce context-specific social stress among individuals, challenging their emotional, mental, and physiological wellbeing as distress.

In the cognitive activation theory of stress (CATS), social stress is better portrayed as a negative cognitive–emotional condition of feeling hopelessness and helplessness when one perceives difficulty in dealing with minor or major life events, as well as daily interpersonal encounters [27]. Accumulating lines of evidences demonstrate that one’s social status is a predominant driver of perceived social stress [21,24,28], which largely determines one’s stable access to societal resources (e.g., employment, life partner, housing) [17,29], ease of managing work-based social interactions [30], and magnitude of cumulative social disadvantages throughout the course of life [31,32,33]. This can be exemplified by a study in China which demonstrated that one’s perceived social status, its change over time, and peer comparison significantly influence one’s psychological wellbeing due to social transition [34]. Similarly, education is often seen as a means to climb the social ladder, which is constructed by socially embedded norms and values in societies, to achieve better social status in Western societies. With attendance at an elite university being a known positive factor contributing to later-life social advantages [31], the notion of the social status–perceived stress paradigm is further supported by the expectation of excelling in university entry public exams (e.g., SATs, GCSEs) [35], suggesting social expectation as a source of social pressure. Furthermore, social dynamics-relevant perceived stress can be caused by one’s dissatisfaction due to self- and other-driven social expectation regarding their social status, work, and even subjective feelings and wellness [30,36,37,38]. These, together, determine the magnitude of cumulative stress (i.e., chronic stress) one experiences in life, as well as the length of time for which one experiences such stress.

### 2.2. Biology of Social Stress and Its Impact on Health of Historically Marginalized Populations

Experiencing a one-time stress event, especially a major adverse life event, and chronic stress have been associated with poorer health outcomes [39,40,41]. Researchers have identified the connection between cumulative stress and a higher incidence of adverse physical, mental, and behavioral health outcomes such as cognitive decline, cardiovascular and inflammatory disorders, and health risk behaviors (e.g., substance abuse) [20,42,43,44]. During evolution, our ancestors passed on the genes that establish the stress response neuroendocrine system, especially the glucocorticoids and hypothalamus–pituitary–adrenal (HPA) axis. This stress response cascade is beneficial to one’s survival in the face of acute threats but is usually harmful to health under long-term stress. While the stress response induced by the HPA axis responds to acute stress promptly and returns to the normal basal level right after, chronic stress was recently found to dysregulate the production of HPA stress hormones by increasing their gland mass, prolonging HPA axis activation and its subsequent detrimental tissue damage [45]. Rodent models of social stress in the form of social defeat (i.e., experiencing social exclusion) have demonstrated that a single stress experience alone is sufficient to elicit long-term effects in the symptomatology of the stress response cascade, mainly via sensitizing one’s stress reactivity toward subsequent stressors [46,47]. This is believed to be partly mediated through the alteration of neuroendocrine reactivity, likely by changing corticosteroid receptor levels for glucocorticoid (GR) and mineralocorticoid (MR) in the brain regions involved in stress reactions (i.e., prefrontal cortex, hippocampus, and hypothalamus) and the modification of neurobiology [46,48,49].

Not surprisingly, socially relevant chronic stress, in animal studies, has been shown to severely dysregulate the HPA axis’ activity [49], impair adult hippocampal neurogenesis [29], exert neurotoxicity on brain physiology [50], aggravate the stress sensitization process, and alter cognitive functions and social behavioral patterns (e.g., help-seeking, social avoidance) [51,52,53]. Meanwhile, chronic stress-induced social avoidance behavior further augments one’s stress susceptibility and worsens their pathophysiological state by withdrawing from sources of potential social support [54]. Research suggests that the pathophysiology of stress induction stems from altered bidirectional communication between the brain and immune system, which involves the converged activation of pro-inflammatory pathways in neurons, endothelia, microglia, and monocytes [51]. This suggests that experiencing chronic social stress facilitates chronic low-grade inflammation throughout the body. Interestingly, the observation of reduced cardiac autonomic function among those experiencing higher cumulative stress in a community study consolidates this chronic stress-induced inflammation paradigm [55], along with the well-established causality between chronic inflammation and cardiovascular autonomic dysfunction among those with chronic diseases [56,57]. Therefore, chronic stress-induced inflammation accounts, at least in part, for disease pathogenesis in those experiencing chronic social stress.

People from historically marginalized backgrounds, especially those with congenital disabilities, tend to experience more early-life social adversity, cumulative disadvantage, and chronic social stress early on in their life [17,58]. This early inequality in societal force-pertaining life experiences causes differentiations in their stress physiology and trajectories of emotional, mental, and physiological wellbeing, which may explain their health disparity over the course of life [28,33,58,59,60]. These historically excluded groups are more likely to suffer from lower subjective social status and objective socioeconomic status due to microaggressions and stigmatizing traits [61] and, thus, experience greater constraints [62] and perceived social stress throughout the course of life [21,22,24]. Early childhood adversity and cumulative disadvantage are both known to elevate stress reactivity (i.e., affective reactivity to daily life stress) and its subsequent perceived cumulative stress over the course of life [21,22,24,60], which can easily lead to allostatic overload and related health inequalities [32,59]. Therefore, the chronic social stress-induced health disparity mostly acts through a pro-inflammatory phenotype [39] and chronic low-grade inflammation, likely developed from the prolonged activation of the HPA axis and dysregulated stress response cascade [40,41,63,64]. In this way, long-term social pressure exerts its effects on health disparities among people who are historically marginalized in terms of social stratification.

### 2.3. Social Support Theory

Social support is defined as the perception of being cared for by interpersonal interaction, either in the form of verbal or nonverbal communication, between recipients and providers that either enhance life satisfaction by diminishing uncertainty in one’s life experience or reduce one’s perceived social stress by modifying stress appraisal (i.e., stress internalizing process) [65,66,67]. During the process, social support enables the recipients to view their life per se as more manageable and thus facilitates their homeostatic state both psychologically and physiologically [66]. Social support is generally classified into five categories—informational, emotional, esteem, social network support, and tangible support [65,68]. Information support is defined as messages that include advice, facts, or feedback on actions [68]. Emotional support refers to expressions that entail caring, concern, empathy, and sympathy [68]. Esteem support includes messages that improve one’s confidence in their skills, abilities, and intrinsic value [68]. Social network support is defined as the messages that elevate one’s sense of belonging to a specific group of people who share similar interests and concerns [68]. Lastly, tangible support entails the provision of any needed physical goods (e.g., money and food) and services (e.g., transportation) to recipients [68].

Social support was first proposed to influence our wellbeing through main-effect and stress-buffering models, according to Cohen and Wills’ paradigmatic definition [66]. Both conceptualizations of social support and its beneficial effects on health have been further studied and developed by sociopsychologists in various terms [69,70,71,72]. For example, the ideas of nurturant support and action-facilitating support were raised by Cutrona and Suhr, who stated that the abovementioned five types of social support work via providing comfort to improve one’s perceived life experience (i.e., altering the stress appraisal process) and solving challenging situations that bring on distress, respectively [68]. Depending on the social context and controllability of stressful events, various types of social support help the recipients’ adaptive coping with stress with varying effectiveness [68,69,70]. In addition to promoting stress-buffering efficiency, social support also exerts direct salutary effects on wellbeing, solely by facilitating positive emotional feelings, life satisfaction, and a “sense of coherence” (SOC) [67,71,73]. In the main-effect model, social support consolidates one’s SOC, in which their view toward the world and one’s environment is more comprehensive, manageable, and meaningful. Subsequently, it enhances one’s perceived social support, which improves affective wellbeing (i.e., more positive affects and less negative affects) and, thus, hedonic wellbeing (i.e., life satisfaction, positive and negative affects). In this way, perceived social support can improve subjective wellbeing, which benefits health and longevity [74]. Social integration allows one access to social network support, which was proved to elevate subjective wellbeing [75]. Collectively, the health-promoting mechanisms of social support in real-life circumstances can be better portrayed in Bailey et al.’s composite model of stress and social support (see Figure 1 in reference [76]), by which perceived support promotes both one’s stress coping efficiency and subjective wellbeing over the course of life. Therefore, social dynamics-induced perceived support is beneficial to psychological, mental, and physiological health [12,77,78].

### 2.4. Biology of Social Support and Its Impact on Health of Historically Marginalized Populations

Experiencing and perceiving social support has been associated with better physical health because it modifies stress physiology and improves subjective wellbeing. Prior research has found that receiving and perceiving high levels of social support is a reliable protective factor against mental illness, chronic diseases (e.g., depression and cancer), and all-cause mortality in both children and elderly people, therefore serving as a good predicator of better physiological health and longevity [19,77,79,80]. Similarly, social capital and social integration (i.e., quantity and quality of social network) protect people from developing chronic diseases (e.g., diabetes, heart disease) [81]. The mechanisms underlying this are not completely understood, but a 2018 meta-analysis study concluded that social support and social integration contribute to lower levels of inflammatory cytokines [82]. The notion that the regulation of the inflammatory response and phenotype largely mediate the social support-pertaining salutary effects is widely acknowledged. Currently, there are five potential psychobiological pathways proposed for the regulatory role of perceived social support on chronic inflammation—adherence to healthy lifestyle [83,84,85], the autonomic nervous system (ANS), the neuroendocrine system, the central nervous system, and the immune system [86].

Social support has been demonstrated to induce better adherence to a healthy lifestyle and elevated life quality [83,84,85], which protect against chronic inflammation [87]. Accumulating lines of evidence validate the linkage between social support and healthy lifestyle behaviors and health outcomes such as better sleep quality [88,89], healthy diet intake [90,91,92], and sticking to a regular physical exercise regimen [84,93], suggesting that social support exerts beneficial effects on the functional integrity of the body. Generally speaking, social support from friends and family facilitates one’s adherence to healthy lifestyle regimens through the provision of encouragement, connection, accountability, and behavior sharing [83]. All of these healthy lifestyle regimens are effective at preventing cognitive impairment [94,95,96], improving executive functioning in both adults with and without the risk of cognitive decline [97,98,99,100,101]. In addition, participating in these healthy lifestyle regimens also ameliorates subjective wellbeing (e.g., happiness, life satisfaction) in both young adults and elderly people [102,103,104]. This further consolidates the cognitive beliefs of sticking to these lifestyle regimens as a positive feedback mechanism [90,105,106,107]. Given the well-known roles of good sleep quality, a healthy diet, and physical exercises in mitigating the harmful effect of chronic inflammation [108,109,110,111], lasting adherence to healthy lifestyle behaviors partly contributes to the anti-inflammatory effects of social support on physiological health.

Perceived social support through supportive social networks promotes the functional balance of the ANS, which promotes inflammation homeostasis [112,113,114]. When the ANS becomes dysregulated by psychological distress, it leads to an autonomic imbalance characterized by sympathetic overactivity and parasympathetic withdrawal, which causes pro-inflammatory phenotypes and favors pathogenesis [115,116]. Receiving and perceiving strong social support has been associated with elevated parasympathetic activity, as reflected by significant high-frequency heart rate variability in people under stress either in the form of stressful circumstances (e.g., suicidal ideation, rumination) or inflammatory disorders (e.g., ulcerative colitis) [112,117,118]. Furthermore, such parasympathetic dominance can activate the cholinergic anti-inflammatory pathway in acetylcholine receptors expressing neurons to suppress cytokine production, thus reducing the propensity for pro-inflammatory phenotypes both systemically and in the brain (e.g., neuroinflammation) [119,120]. However, experiencing early-life trauma, a form of social stress, likely alters psychophysiology and the regulatory function of the ANS, and without timely resolution, early-life trauma could even progress into post-traumatic stress disorder (PTSD) later in life [121]. PTSD chronically hinders the ANS from achieving the “sweet spot” of balancing autonomic states in various ways [122], leading to prolonged and unwanted stress responses in the absence of social stressors. While the aging process aggravates the aforementioned autonomic imbalance, perceived social support-related parasympathetic dominance is found to promote health and longevity [86,123,124]. This mechanism underlying social support-driven salutary and stress-buffering effects can be explained by two influential theories—Porges’s polyvagal theory [124] and Thayer’s neurovisceral integration theory [123]. In short, both theories posit that perceived safe and supportive environments facilitate increased parasympathetic control by the mammalian myelinated vagus nerve [123,124], thereby reducing sympathetic stress reactivity and enhancing positive social behavior [125,126]. In this way, the fight-or-flight response is inhibited while the heartbeat and activity of the HPA axis are slowed, resulting in the modulation of the immune system and an attenuated inflammation profile [86]. By doing so, social support-induced parasympathetic dominance exerts anti-inflammatory effects.

Additionally, social support can dampen the HPA axis’ activity and stress reactivity through modulating the neuroendocrine system for its partial salutary effects. The oxytocin system is the most prominent neuroendocrine system that is responsible for the stress-buffering effects of social support [127,128,129,130]. During supportive social interactions, oxytocin is released into the bloodstream by the posterior pituitary in response to sensory stimuli (e.g., touch, warmth), mental images, and emotional states that are perceived as positive (e.g., feeling loved) [131,132,133,134]. First, oxytocin downregulates the HPA axis by activating hippocampal GABAergic neurons [135], and it subsequently inhibits corticotropin-releasing factor (CRF) expression at the paraventricular nucleus of the hypothalamus [136]. Oxytocin also attenuates the “fight-or-flight” response-relevant noradrenaline release in the locus coeruleus and nucleus tractus solitarius in the brain [137]. This leads to an attenuated stress response [138]. Furthermore, oxytocin exerts stress-buffering effects by diminishing glucocorticoid levels (e.g., cortisol) [128,134], suppressing cardiovascular stress responses [139], reducing amygdala responsivity to stress [140], and modulating emotion-related opioidergic and serotoninergic activities in the brain [141]. All these findings are in concordance with the previously reported anti-inflammatory properties of oxytocin release [142].

Lastly, perceived social support alters the central nervous system and immune system, which help buffer stress physiology. Receiving social support on a daily basis can positively modulate the neural mechanisms that facilitate the learning process of interpreting social interaction as rewarding and the unlearning process of stress reactivity in response to negative emotional stimuli (e.g., fear and pain) [143]. This may be partly mediated by reduced activity in the dorsal anterior cingulate cortex (dACC) and Brodmann’s area 8 in the dorsal superior gyrus [144], the brain regions previously associated with dealing with distressing social experiences/neurocognitive reactivity to social stressors (e.g., social separation) [145,146]. By desensitizing the dACC through the release of opioids over time, reduced neuroendocrine stress responses can be achieved among those receiving social support [144]. In parallel, perceived social support is reported to boost immunocompetence against inflammation by augmenting the sensitivity of immune cells to the anti-inflammatory actions of glucocorticoids [147]. This salutary effect is thought to be mediated by the increased expression of the glucocorticoid receptor gene and lower expression of genes that respond to the pro-inflammatory transcription factor NF-kappa B [147]. The improvement in immune response upon social support is also reflected by better natural killer cell numbers and activity (i.e., better immune function) among well-supported individuals [80,148]. Taken together, this concludes the multiple beneficial mechanisms of social support, which help to soothe stress pathophysiology, ease stress reactivity, and improve one’s stress coping capacity. Therefore, perceived social support is effective in protecting against the social stress-induced inflammatory phenotype.

People from socially disadvantaged backgrounds, especially those with disabilities, likely receive much less social support than those with social privilege (i.e., the benefits, opportunities, and a lack of social difficulties experienced by members of socially dominant groups in society, e.g., the white racial group) [149,150,151,152]. Statistics and research indicate that they usually suffer from inequality in social capital (e.g., friends) and more social isolation [149,150,151], thus restricting their potential sources of high-quality social support amidst challenges. Notably, people from these marginalized communities tend to experience higher levels of trauma in the forms of direct neglect, isolation, and discrimination, thus likely developing a greater extent of societal mistrust and influencing their perceptions of social support [153,154,155,156,157]. This partly explains the observed lower perceived social support and subsequent allostatic overload among them. With respect to the high likelihood of allostatic overload in their physiological health, these historically marginalized persons necessitate social support to effectively compensate for some of the detrimental health damage brought by social pressure. However, perceived social stress has a more significant influence over perceived social support when it comes to shaping pro- or anti-inflammatory phenotypes among populations who encounter constant social pressure [158]. Fortunately, a growing body of recent evidence highlights the pivotal role of social support in developing their resilience against physiological stressors [159,160]. With more high-quality social support, these historically marginalized populations will likely improve their regulatory capacity toward stress through remodeling the neural landscape of stress physiology. For instance, oxytocin remodels the neuronal networks associated with stress appraisal, emotions, and social interaction [161], which enhance the learning of stress reappraisal and support-seeking processes via pro-social behaviors [130,162]. Moreover, social support can facilitate life satisfaction by meeting basic psychological needs (e.g., connection, love) and stabilizing emotional states (e.g., peace of mind), thus improving psychological resilience [160]. Overall, the combined effects of perceived social support and its associated neuroendocrine adjustment prevent the HPA axis and noradrenergic activity from fluctuating out of the optimal scale during daily life stress [125,127,128,133,134,138,159]. Therefore, this minimizes the long-term influence of social stress on psychological, mental, and physiological health among marginalized individuals.

### 2.5. The Biological Effects of Social Dynamics on the Aging Trajectories of Historically Marginalized Individuals

According to the described biology of social stress and social support, social dynamics have multidimensional influences on various facets of one’s psychological, emotional, mental, and physiological wellbeing, which are intertwined in a crosstalk bidirectionally. To comprehensively interpret the temporal effect of social dynamics on aging trajectory, adopting the life-course approach is optimal for dissecting this multidimensionality.

Prominently, social dynamics influence the aging process directly and indirectly as one of the environmental risk factors in one’s exposome (see Figure 1). The nature of one’s social dynamics, at least in part, determines the quality of one’s interpersonal interactions, the size of one’s social network, and the scale of social capital, which all determine the quantity and quality of social support one can receive. Meanwhile, one’s subjective social standing and objective socioeconomic status, as well as one’s quantity of interpersonal conflicts [25], largely govern the magnitude of social stress experienced in daily life. Based on the idea of the exposome, the amount of social support one receives can compensate, partly, for the psychological distress caused by the social stress one experiences every day, meaning that it is the predominant factor determining the cumulative effect of social dynamics on one’s stress physiology, subjective wellbeing (e.g., life satisfaction), and, thus, their rate of biological aging. As a result, everyone has unique personal life experiences resulting from their heterogenous pattern of social dynamics, leading to heterogeneity in health. Additionally, social dynamics have been found to affect one’s all-cause mortality and, thus, lifespan through the degree to which one experiences social integration and social isolation [163]. Numerous studies have demonstrated that social integration reduces the risk of all-cause mortality [12], while social isolation increases the risk [18]. The biological effects of social integration and social isolation on lifespan are mediated by inflammation, according to a number of recent population studies [11,20,164]. Therefore, the role of social dynamics in modulating the aging process is largely regulated by their effects on inflammation.

Inflammation is the central hub of biological aging as its outcome and driver. The bidirectional causality of biological aging and inflammation in their interconnected relationship has been validated by a large body of proof-of-concept epidemiological [165,166,167], animal [168,169], and cell [170,171] studies. During the aging process, the temporal accumulation of cellular and tissue damage, as well as immune dysfunction, gradually disrupts the homeostatic balance between pro-inflammatory and anti-inflammatory signaling, resulting in low-grade systemic chronic inflammation [171,172]. Under such persistent inflammation, a heightened inflammation profile further accelerates the rate of biological aging through aggravating the dysfunction of other molecular hallmarks of aging, including cellular senescence in cells from various vital organs and the immune system (i.e., immunosenescence) [170,171,172,173,174,175,176]. In this process of inflammaging, the dysregulated pro-inflammatory phenotypes significantly facilitate the pathogenesis of age-related chronic diseases [177], especially neurocognitive functional decline [178]. Thus, inflammation-associated neurocognitive alteration likely hinders the cognitive stress appraisal process, resulting in a higher magnitude of perceived stress and its consequential elevated activity in stress pathophysiology [179,180]. Therefore, psychosocial drivers of inflammation accelerate the rate of cognitive and physiological decline and, thus, the rate of biological aging, while its psychosocial inhibitors do the opposite.

Apart from the direct biological effects of social stress and social support brought by social dynamics, one’s aging trajectory is also influenced by the cumulative net outcome of their quality of life (QoL) every day, resulting from the multiple dimensions of the exposome, including social dynamics. QoL is the subjective rating of one’s life experience; life can be perceived as desirable or undesirable at a specific time point based on the psychological, physical, social, and emotional domains of wellbeing [181,182]. QoL indirectly mediates the severity of inflammation, with there being an inverse relationship between these factors [183,184,185]. One’s subjective wellbeing (QoL), the relative severity of one’s inflammation state, and one’s social behaviors are all mutually regulated, as illustrated with the following two examples. Firstly, people who receive high-quality perceived social support usually have better cognitive health and performance [186]. As a result, the stress reappraisal and stress coping capacities of these people improve [180], thus lowering the severity of inflammation and its detrimental effects on emotion (e.g., regulation and recognition) [23] and lifestyle (e.g., ability of exercising) [187]. This eventually improves their emotional, mental, physical, and subjective wellbeing. As another example, people who suffer from inflammation experience neurobiological alterations in emotional attention that increase their sensitivity toward negative social dynamics (i.e., stress reactivity) and positive social encounters (i.e., reward recognition) [23,188]. Depending on the context of social dynamics, these individuals adopt either social withdrawal or social approach behaviors [189]. Based on the availability of a supportive community and the nature of social attitudes, they can have differences in QoL and subsequent inflammation modulation but likely suffer from lower QoL and depression [10]. This is in line with the previous community-based findings that chronic inflammation is accompanied by lower QoL [190,191]. These two examples indicate that positive social dynamics fulfill basic psychological needs and help one adhere to a physical health regimen [91,93], reach peace of mind, and achieve social integration, whereas negative ones do the opposite. This also suggests that subjective wellbeing and, by extension, the cumulative biological effects of social dynamics that one experiences over the course of life are unfolded in their inflammation profiles. Therefore, social dynamics have complex multilateral psychobiological effects on one’s psychological, physical, emotional, and social needs, which determine the wholeness of one’s wellbeing.

Altogether, social dynamics can modulate, by either aggravating or soothing, the rate of wear and tear/physiological decline and, thus, the rate of maintaining the inflammation homeostasis state that normally deteriorates over time. In this way, social dynamics, depending on contexts and quality, can regulate the timing of the onset and overall time of chronic low-grade inflammation over the course of life and, thus, determine the rate of biological aging in individuals. According to the overarching biopsychosocial model, the biological effects of social dynamics, under the framework of perceived social stress and social support, on the biological aging of a historically marginalized individual are briefly illustrated in Figure 3. Therefore, the psychosocial aspect of biological aging and the later-life risk of getting chronic diseases can be malleated through one’s social dynamics and life experience.

## 3. Discussion and Future Directions

To our knowledge, this review is the first to demonstrate and summarize the interconnection between social dynamics and the biological aging of historically marginalized individuals through a transdisciplinary overarching model intersecting sociology, psychology, neuroscience, environmental science, and biogerontology under the solid framework of social stress and social support theories. Since the current health disparity research rarely addresses an interdisciplinary framework linking societal forces and health inequality, we propose this overarching model to lay out the theoretical framework of the potential biopsychosocial mechanisms underlying the health disparity observed in most historically excluded populations. Existing public health research has identified the association of social exclusion and higher morbidity risk in populations who are socially marginalized in terms of race, ethnicity, and immigration status [17,192]. However, the mechanistic basis underlying the health disparity observed in these marginalized populations remains less understood. The overarching paradigm proposed in this review may explain the mechanisms underlying the prior findings about the health challenges in various socially excluded populations and will inform future intervention strategies in addressing health disparities.

To further elucidate this biopsychosocial model, measuring inflammatory biomarkers (e.g., interleukin 6) and DNA methylation-based biomarkers [193] in non-invasive bio-samples (e.g., saliva and plasma) and neuroimaging (e.g., EEG) will be needed in future epidemiological studies. Studies should be in the form of retrospective and prospective longitudinal research focused on people from historically marginalized communities. This would ensure that outstanding questions surrounding the influence of social dynamics on biological aging can be tested to better formulate institutional policies and develop programs to further reduce social stress and facilitate social support. Understanding how community programs aimed at improving life circumstances (e.g., high-quality foods, clean indoor air) modulate the levels of inflammatory biomarkers and epigenetic age in marginalized ethnic and immigrant groups perfectly exemplifies the transformative impact of the biopsychosocial model proposed in this review.

We believe this review serves as a timely paradigm that can help to fill the knowledge gaps regarding the psychosocial drivers of biological aging and the biology of negative social dynamics (i.e., social discrimination) in marginalized populations, especially in light of the emerging interest in national funding agencies. This past year, the National Institute on Minority Health and Health Disparities (NIMHD) announced that they will recognize people with disabilities as disparity populations for funding purposes in response to prior questioning of the status quo [150], reflecting a pivotal recognition of the marginalization of such populations by national funding agencies. Moreover, the National Institute of Aging (NIA) has identified understanding the effects of social dynamics on aging as one of nine goals listed in their 2020–2025 strategic research directions. In this vein, studying the effects of ableism, including audism, on the health of people with disabilities will facilitate our understanding of the psychobiological impact of social dynamics on broader society as well as its impact on people with various personal characteristics and corresponding aging trajectories. Therefore, this review could provide the model framework for future research studies on the health disparities of historically marginalized groups, such as Deaf and hard-of-hearing populations.

## 4. Concluding Remarks

Social dynamics are closely related to our wellbeing. However, the role of social dynamics as a significant social determinant of health, especially their impact on biological aging in historically marginalized populations, has not been fully addressed. Based on emerging scientific evidence across disciplines, we have proposed an overarching biopsychosocial model, which helped to elucidate that social dynamics can influence the rate of biological aging and, consequently, induce health disparities in these marginalized populations. Through a theoretical framework of social stress and social support in this overarching model, we have briefly summarized the existing mechanistic understanding about the psychobiological effects underlying the social dynamics of historically marginalized populations. As social dynamics can be particularly challenging to historically excluded populations, including people with disabilities, our overarching model will serve as an interdisciplinary and holistic paradigm for future research efforts aimed at understanding and addressing the psychosocial impacts of social dynamics on their health, wellbeing, biological aging, and health disparities.

## Figures and Tables

**Figure 1 ijerph-21-00554-f001:**
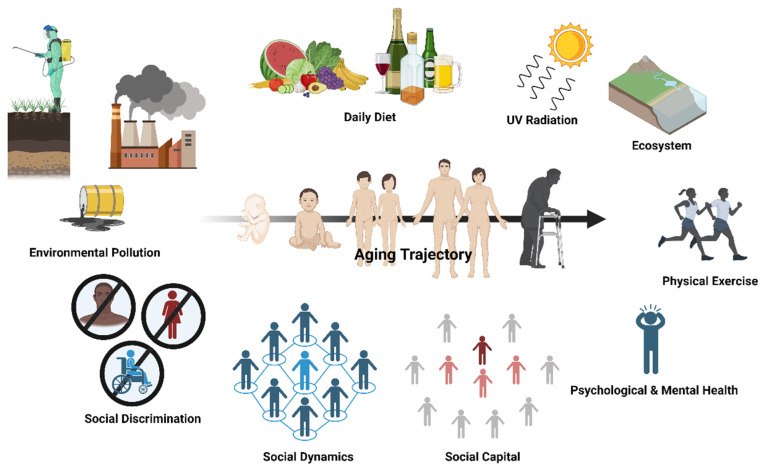
One’s aging trajectory is malleable and can be formed through various facets of the exposome, with social dynamics being one of the social determinants of health. The aging trajectory of an individual depends on a multitude of environmental factors and lifestyle behaviors. This includes environmental pollution, social discrimination, social dynamics, social capital, psychological and mental health, physical exercise, diet, irradiation, and our interaction with the physical environment (i.e., ecosystem). Social dynamics are multidimensional social determinants of health that are bidirectionally related with social discrimination and social capital, which all have an impact on healthy lifestyle adherence (e.g., daily diet, physical exercise regime) and psychological and mental health. Created with Biorender.com.

**Figure 2 ijerph-21-00554-f002:**
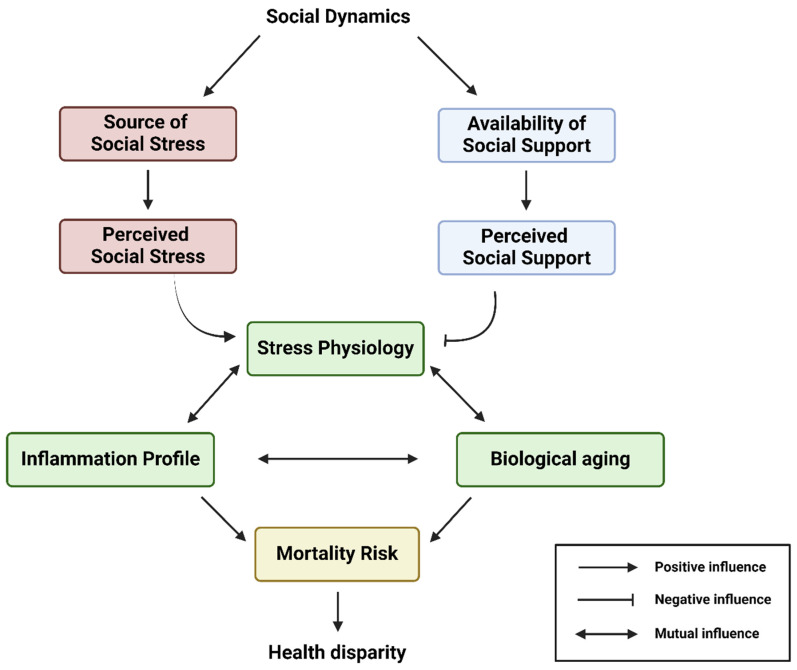
This conceptual flow chart illustrates the overarching biopsychosocial model interconnecting the social dynamics, availability of social support, source of social stress, stress physiology, inflammation, biological aging, and health disparity of an individual. Social dynamics largely determine the source of social stress and availability of social support networks. In this model, a historically marginalized individual can experience various degrees of perceived social stress and perceived social support, which drive their stress physiology, inflammation profile, and biological aging. Eventually, all these will sculpt the range of mortality risk and, thus, the life course health disparity outcome of the historically marginalized individual. Created with Biorender.com.

**Figure 3 ijerph-21-00554-f003:**
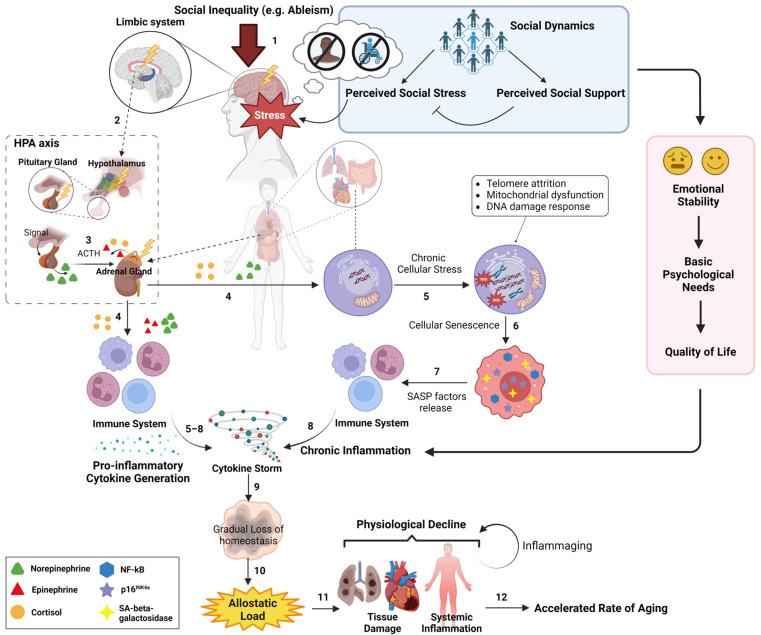
A graphic representation of the biopsychosocial model that illustrates the sequential order of social inequality-triggered physiological influences on the aging process. On a daily scale, an individual from historically marginalized populations (e.g., people with disabilities, historically marginalized racial groups) experiences their unique social dynamics, where the magnitude of their physiological stress (shown as “Stress”) is determined by the net outcome of their perceived social stress and perceived social support (see blue box) (1). This physiological stress subsequently triggers the HPA axis from the limbic system (2), which initiates the release of neuroendocrine hormones (e.g., cortisol) (3). Over time, these neuroendocrine hormones, in turn, induce chronic cellular stress in various tissues (e.g., lung) and immune cells from immune system (4), resulting in pro-inflammatory phenotypes (e.g., cytokine storm) (5–8). These pro-inflammatory phenotypes further aggravate the deterioration of tissue homeostasis upon the wear and tear damage during aging (9). In parallel, social dynamics affect one’s emotional stability, basic psychological needs, and quality of life (see red box), all of which can modulate the profile of chronic inflammation. Altogether, social dynamics can directly and indirectly influence the inflammation profile, allostatic load (10), and the rate of physiological decline (11), eventually modulating the rate of biological aging (12). The blue box represents the sociopsychological aspect of social dynamics and its effect on determining the cumulative physiological stress of the individual, starting from the HPA axis. The red box represents the effect of social dynamics on subjective wellbeing, which modulates the inflammation profile. Created with Biorender.com.

## Data Availability

Not applicable.

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
