# Peer review of "The Influence of Social Dynamics on Biological Aging and the Health of Historically Marginalized Populations: A Biopsychosocial Model for Health Disparities"

_ijerph, 2024, doi:10.3390/ijerph21050554_

Round 1
Reviewer 1 Report
Comments and Suggestions for Authors
This paper reviews the factors that increase or mitigate social stress to historically minoritized populations and considers how this social stress leads to physiological changes that increase biological aging and lead to poor health outcomes. The paper discusses the role of a wide range of physiological mechanisms, demonstrating how they can be triggered by social stress or ameliorated by perceived social support, generating a highly plausible biopsychosocial model of the consequences of social inequality on the aging process.
This paper provides for the first time an overarching model that links together concepts of social support, social stress and social inequality experienced by marginalized populations, with known biological mechanisms from the fields of neuroscience, immunology and gerontology that are likely to be underpinning the considerable health inequalities observed within these populations. The paper was very well written and a delight to read, with a logical flow that well described a complex biopsychosocial interaction. While I found myself with questions as I was reading the paper, it was pleasing to see that most were quickly answered as I progressed through the paper. This new conceptual model will be an excellent contribution to the literature in understanding what drives health inequalities in marginalized populations.
I have two minor points that I would like to see addressed in the manuscript:
1) In the paragraph starting on line 286 about how social support impacts ANS balance, while this was reasonably comprehensive, an aspect that was missing was the role of trauma on the ANS and how trauma interacts with social support and social stress. This is particularly important in marginalised communities, such as people with disability, who as a population experience higher levels of trauma, either as direct acts of abuse or neglect, or through discrimination, isolation and ableism. I would like to see mention of trauma as a type of social stress and how this can have in impact on the ANS. The mention of the polyvagal theory within that paragraph already sets this concept up, but it needs to be explicitly discussed given that early trauma can have a lifetime of effects.
2) In the Discussion and Future Directions section, while I agree that it is extremely positive to see that funding is now being prioritised for research on minority populations such as people with disability, and that the effect of social dynamics on aging is also a research priority, these are not actual ‘future directions’ when it comes to understanding or validating the model proposed within this paper. I would like to see the authors discuss some possible research strategies for testing their proposed model. While the model is logically plausible, it is a new model and hypotheses need to be testable so the authors should make some suggestions on how this could be done. For instance, what interventions to improve social support or otherwise decrease social stress could be used to test whether biological aging could be slowed, and how would that be measured in a relevant minority population? Or is there some sort of test (whether via clinical study or longitudinal survey) that could be applied on a population level (or large sample) to determine if relevant minority populations who are experiencing social stress have some of the proposed neuroimmunological changes that may be increasing aging-related disease or other markers of biological aging?
Author Response
Reviewer 1:
This paper reviews the factors that increase or mitigate social stress to historically minoritized populations and considers how this social stress leads to physiological changes that increase biological aging and lead to poor health outcomes. The paper discusses the role of a wide range of physiological mechanisms, demonstrating how they can be triggered by social stress or ameliorated by perceived social support, generating a highly plausible biopsychosocial model of the consequences of social inequality on the aging process.
This paper provides for the first time an overarching model that links together concepts of social support, social stress and social inequality experienced by marginalized populations, with known biological mechanisms from the fields of neuroscience, immunology and gerontology that are likely to be underpinning the considerable health inequalities observed within these populations. The paper was very well written and a delight to read, with a logical flow that well described a complex biopsychosocial interaction. While I found myself with questions as I was reading the paper, it was pleasing to see that most were quickly answered as I progressed through the paper. This new conceptual model will be an excellent contribution to the literature in understanding what drives health inequalities in marginalized populations.
We appreciate the reviewer 1’s feedback and recognition of the significance of this manuscript in driving our understanding of health disparities in marginalized populations.
I have two minor points that I would like to see addressed in the manuscript:
1) In the paragraph starting on line 286 about how social support impacts ANS balance, while this was reasonably comprehensive, an aspect that was missing was the role of trauma on the ANS and how trauma interacts with social support and social stress. This is particularly important in marginalised communities, such as people with disability, who as a population experience higher levels of trauma, either as direct acts of abuse or neglect, or through discrimination, isolation and ableism. I would like to see mention of trauma as a type of social stress and how this can have in impact on the ANS. The mention of the polyvagal theory within that paragraph already sets this concept up, but it needs to be explicitly discussed given that early trauma can have a lifetime of effects.
We agree with this reviewer that trauma and PTSD can play a modulating role of the ANS in the stress psychophysiology and thus influence the stress response in those with trauma experience. To address this, we have introduced the potential impact of early life trauma in psychophysiology in the paragraph of ANS (line 307-313) for the biology part and mentioned that people from historically marginalized social background tend to experience more trauma and thus become less likely to perceive social support (in line 368-372).
2) In the Discussion and Future Directions section, while I agree that it is extremely positive to see that funding is now being prioritised for research on minority populations such as people with disability, and that the effect of social dynamics on aging is also a research priority, these are not actual ‘future directions’ when it comes to understanding or validating the model proposed within this paper. I would like to see the authors discuss some possible research strategies for testing their proposed model. While the model is logically plausible, it is a new model and hypotheses need to be testable so the authors should make some suggestions on how this could be done. For instance, what interventions to improve social support or otherwise decrease social stress could be used to test whether biological aging could be slowed, and how would that be measured in a relevant minority population? Or is there some sort of test (whether via clinical study or longitudinal survey) that could be applied on a population level (or large sample) to determine if relevant minority populations who are experiencing social stress have some of the proposed neuroimmunological changes that may be increasing aging-related disease or other markers of biological aging?
We agree with this reviewer that recommendation of research strategies to test the biopsychosocial model proposed is needed in the discussion and future directions part. We have supplemented an extra paragraph (in line 513-523) to explain the current research logistics required for understanding the social dynamics-biological aging-health disparity axis by suggesting biomarker and neuroimaging research method in future epidemiological studies, as well as providing an example of testing this overarching model.
Reviewer 2 Report
Comments and Suggestions for Authors
This is a fascinating and potentially important summary of emerging issues in biological aging and the influence of social factors. I think it is well presented and based on a seemingly comprehensive review of relevant literature. I have two concerns.
1) You have used the phrase "minoritized populations" and sometimes indicate that you see persons with disabilities as a prime example. I do not know what is intended by "minoritized" means and it needs explanation. This lack of clarity leads to confusing claims about social supports available throughout the lifecourse that may apply to persons with disabilities but seem irrelevant to persons from historically oppressed racial/ethnic groups in specific countries. By grouping all groups with less than ideal average outcomes into a single category, you miss much of the complexity around the social dynamics of racism, sexism, homophobia etc. Many readers could see this single category as ahistorical and insulting if it does or does not include their groups and their specific cultural strengthsn and challenges.
2) While I think it is possible that there is a single, albeit complex, set of pathways between social oppression and health outcomes, it seems much more likely that objective differences in life circumstances (quality of food, air pollution exposure, physical conflict with other groups etc.) create specific biological and psychological stresses that are not mediated by psychological responses. Experiences of famine, brutality, lack of political voice etc. are reasonably associated with life outcomes beyond their impacts on psychological elements of aging.
3) You identify a number of specific psychophysiological pathways. Are you claiming that these are the only, or most important pathways? I think you should indicate what other pathways may be relevant and why you discount these.
Author Response
Reviewer 2:
This is a fascinating and potentially important summary of emerging issues in biological aging and the influence of social factors. I think it is well presented and based on a seemingly comprehensive review of relevant literature. I have two concerns.
Thank you for your comments and recognition.
1) you have used the phrase "minoritized populations" and sometimes indicate that you see persons with disabilities as a prime example. I do not know what is intended by "minoritized" means and it needs explanation. This lack of clarity leads to confusing claims about social supports available throughout the lifecourse that may apply to persons with disabilities but seem irrelevant to persons from historically oppressed racial/ethnic groups in specific countries. By grouping all groups with less than ideal average outcomes into a single category, you miss much of the complexity around the social dynamics of racism, sexism, homophobia etc. Many readers could see this single category as ahistorical and insulting if it does or does not include their groups and their specific cultural strengthsn and challenges.
We agree with this reviewer for the confusion caused by “minoritized populations” in the manuscript. To address this, we have defined the historically minoritized populations in the introduction (in lines 92-94) and explain why these populations are likely to suffer from higher level of social stress overall. It is true that these groups do might not have the same experiences and struggles, but when it comes to the social dynamics and biological aging concepts discussed in this review, there is application relevance to all of the groups. For example, there is intersectionality within people with disabilities who meanwhile can belong to specific racial minority groups (e.g., Asian, black, brown), different sexual orientation (e.g., LGBTQ+) as well as have different immigration and socioeconomic status. This helps frame the readers’ thinking to categorize the social factors that impose social stress and social support in each social dynamics as a template framework. We are, by no means, to ignore the acknowledgement of the complexity around the social dynamics of racism, sexism, homophobia. To further clarify such confusion, we have reinstated our scope of the review and embraced the complexity of every group’s unique experience and challenges (in lines 94-100).
2) While I think it is possible that there is a single, albeit complex, set of pathways between social oppression and health outcomes, it seems much more likely that objective differences in life circumstances (quality of food, air pollution exposure, physical conflict with other groups etc.) create specific biological and psychological stresses that are not mediated by psychological responses. Experiences of famine, brutality, lack of political voice etc. are reasonably associated with life outcomes beyond their impacts on psychological elements of aging.
Yes, we agree with this reviewer that objective differences in life circumstances might overlap with some of the pathways we mentioned in the stress physiology. But it is out of scope of this review. We have clarified that by elucidating the scope of this review in lines 94-100. Also, we discuss studying how improving these objective differences in life circumstances in the future epidemiological studies of minoritized populations to test this model in the future directions part (in lines 520-523).
3) You identify a number of specific psychophysiological pathways. Are you claiming that these are the only, or most important pathways? I think you should indicate what other pathways may be relevant and why you discount these.
We are claiming that these are the most significant pathways but not the only ones, as inflammation is for now one of the most prevalent and widely accepted interpretations of mediating the cellular/tissue physiology into systemic phenotype outcome to drive biological aging in the biogerontology field. However, the effects of any undiscovered and underrated pathways on inflammation remain to be uncovered and elucidated in the future research studies.
Again, we would like to thank the reviewers for the insight and time they took to review our manuscript. We hope that these revisions address their concerns and look forward to hearing from you about the acceptance of this manuscript for publication in IJERPH.
Round 2
Reviewer 2 Report
Comments and Suggestions for Authors
This remains a strong paper.
I do not understand the third sentence in the addition around line 92. I do not see that it responds to the concerns with the term "minoritized" ----I did not understand the paper as offering guidelines. I think your intent is saying that you are outling a framework for addressing how social oppression gets into the body and that it does not attend to the unique experiences of different excluded populations and persons..... still think "minoritized" is problematic: "minority" either references population relative size or acheiving the age of consent....it's kind of a nice way of talking about how oppression works for persons with disabilities and clearly does not fit well for women or large racial/ethnic groups in the US. Why not use, "historically excluded "or "oppressed"?
And please recall, that many BIPOC people find the "minoritized" phrase deeply offensive. Its use sanitizes processes such as enslavement of Africans or the virtual genocides of native persons in the US.....I think the phrase is somehow pseudo-scientific---as freeing the scientist from any positionality relative to the targets of oppression.
Or, why not be explicit that you are speaking about the social oppression of people with disabilities as an example that you think illustrates a general process?
Author Response
Dear Reviewer,
Reviewer #2:
This remains a strong paper.
Thank you for the reviewer’s response and positive feedback.
I do not understand the third sentence in the addition around line 92. I do not see that it responds to the concerns with the term "minoritized" ----I did not understand the paper as offering guidelines. I think your intent is saying that you are outling a framework for addressing how social oppression gets into the body and that it does not attend to the unique experiences of different excluded populations and persons....I still think "minoritized" is problematic: "minority" either references population relative size or acheiving the age of consent....it's kind of a nice way of talking about how oppression works for persons with disabilities and clearly does not fit well for women or large racial/ethnic groups in the US. Why not use, "historically excluded "or "oppressed"?
We agree with the reviewer’s critique regarding the use of “minoritized.” We have changed all the terms “minoritized” to “marginalized” throughout the manuscript and the title of the manuscript as the term “marginalized” could reveal the population is oppressed and excluded from the larger society. We have further elucidated the third sentence around line 92 (line 94-97) to clarify the meaning further.
And please recall, that many BIPOC people find the "minoritized" phrase deeply offensive. Its use sanitizes processes such as enslavement of Africans or the virtual genocides of native persons in the US... I think the phrase is somehow pseudo-scientific---as freeing the scientist from any positionality relative to the targets of oppression.
Or, why not be explicit that you are speaking about the social oppression of people with disabilities as an example that you think illustrates a general process?
We agree with the reviewer’s perspective on our use of the term “minoritized” and we have made the corresponding change to “marginalized”. To clarify, the scientific contents of this manuscript are supported by empirical evidence from Black American studies and disability studies, as well as animal studies studying social stress (e.g., social defects) and social support. Therefore, we believe that “historically marginalized” should better reflect these collective groups’ social experience of being historically marginalized and thus excluded from the society.
Again, we would like to thank the reviewers for the insight and time they took to review our manuscript. We hope that these revisions address their concerns and look forward to hearing from you about the acceptance of this manuscript for publication in IJERPH.